# Re-Evaluating Biologic Pharmacotherapies That Target the Host Response during Sepsis

**DOI:** 10.3390/ijms20236049

**Published:** 2019-11-30

**Authors:** Kristopher M. Tuttle, Matthew D. McDonald, Ethan J. Anderson

**Affiliations:** 1Department of Pharmaceutical Sciences & Experimental Therapeutics, College of Pharmacy; University of Iowa, Iowa City, IA 52242, USA; kris-tuttle@uiowa.edu (K.M.T.); matthew-mcdonald@uiowa.edu (M.D.M.); 2Fraternal Order of Eagles Diabetes Research Center, University of Iowa, Iowa City, IA 52242, USA

**Keywords:** sepsis, pharmacology, biologics, multiple organ dysfunction syndrome, inflammation, coagulation

## Abstract

Multiple organ dysfunction syndrome (MODS) caused by the systemic inflammatory response during sepsis is responsible for millions of deaths worldwide each year, and despite broad consensus concerning its pathophysiology, no specific or effective therapies exist. Recent efforts to treat and/or prevent MODS have included a variety of biologics, recombinant proteins targeting various components of the host response to the infection (e.g., inflammation, coagulation, etc.) Improvements in molecular biology and pharmaceutical engineering have enabled a wide range of utility for biologics to target various aspects of the systemic inflammatory response. The majority of clinical trials to date have failed to show clinical benefit, but some have demonstrated promising results in certain patient populations. In this review we summarize the underlying rationale and outcome of major clinical trials where biologics have been tested as a pharmacotherapy for MODS in sepsis. A brief description of the study design and overall outcome for each of the major trials are presented. Emphasis is placed on discussing targets and/or trials where promising results were observed. Post hoc analyses of trials where therapy demonstrated harm or additional risk to certain patient subgroups are highlighted, and details are provided about specific trials where more stringent inclusion/exclusion criteria are warranted.

## 1. Introduction

Sepsis has recently been defined clinically as “a life-threatening organ dysfunction caused by a dysregulated host response to infection” [1]. While standard of care generally focuses on antimicrobial and supportive therapies [2], the effects of acute inflammation in the host (i.e., the patient) must also be addressed. The acute inflammatory component of sepsis, often referred to as systemic inflammatory response syndrome (SIRS), involves the coordinated release of cyto/chemokines, arachidonic acid metabolites, complement, and many other proteins, typically from liver and immune cells. A similar response occurs in many types of injuries, including traumas and burns [3]. Currently, sepsis accounts for >750,000 intensive care unit admissions annually and has been reported to be rising [4]. Despite major advances in emergency medicine in recent decades, mortality rates with sepsis still remain high, reaching greater than 30% if patients develop multiple organ dysfunction syndrome (MODS) [5,6]. Thus, there is an urgent need to identify new drug targets and develop new pharmacotherapies directed at those targets, which can halt the development of MODS.

Recombinant proteins (i.e., ‘biologics’) have become a prominent class of medications to combat both rare and common conditions. They represent a new era of promising medications, and unlike small molecules, biologics can be engineered to serve as decoys, mimics, or direct inhibitors of many types of proteins, including cyto/chemokines, growth factors, immunoglobulins, and enzymes. Recent efforts to use biologics in conjunction with standard-of-care approaches have seen mixed results, but several have shown promise. Major unique challenges exist with sepsis trials, specifically with respect to identifying who to treat and when to treat them (i.e., the therapeutic ‘window’), the highly variable source of the infection, and the presence of confounding co-morbidities in many patients [2].

This review summarizes the underlying rationale and outcome of major clinical trials where biologics were tested as a pharmacotherapy for MODS in sepsis. Trials discussed are grouped by target (Figure 1), and a brief description of the study design and overall outcome are presented for each of the major trials. Targets and/or trials that show promise are emphasized in greater detail than those that clearly failed to show favorable effects. A summary of these trials and their outcomes, along with the current status of each biologic class that is discussed, is provided in Table 1.

## 2. Targeting Cyto/Chemokines

Inflammation in sepsis begins when the host’s innate immune response becomes activated, primarily via toll-like receptors (TLRs). TLRs activate a plethora of signaling cascades, which ultimately results in the generation of numerous inflammatory cytokines and chemokines, the most-characterized of which include tumor necrosis factor alpha (TNFα), interleukins (ILs), and others [7]. Upon activation, TLRs (among others) induce the rapid synthesis and secretion of IL-1β which acts as a positive feedback signal to re-enforce and increase the systemic inflammatory response. IL-1β also works in conjunction with TNFα to increase blood-brain barrier (BBB) permeability. One report documented an association between serum IL-1β levels and risk of mortality [8]. Other studies have reported an association between high levels of IL-6 and risk of adverse outcomes in sepsis [9,10,11]. Randomized clinical trials (RCTs) using biologics to target the cyto/chemokine cascade during sepsis are summarized in the following sections.

### 2.1. Anti-Interleukin Therapy

Interleukin-1 receptor antagonist (IL-1ra) is an endogenous protein produced by immune cells which antagonizes IL-1 mediated responses by competitive inhibition of IL-1 receptors [12,13]. In a pilot study, investigators used human recombinant IL-1ra (hrIL-1ra) in an open-label, placebo-controlled, phase II multicenter trial, to treat sepsis syndrome/shock in 99 patients [14]. A dose-dependent, 28-day survival benefit was associated with hrIL-1ra treatment (*p* = 0.015). Additionally, dose-related survival benefits were observed with infusion of hrIL-1ra in patients with septic shock (*p* = 0.002), Gram-negative infection (*p* = 0.04), and high circulating IL-6 (>100 pg/mL) (*p* = 0.009). In a follow-on study with a much larger cohort, hrIL-1ra treatment did not show significant increases in survival overall, but secondary analyses showed that hrIL-1ra treatment was associated with increased survival time among patients with MODS [15].

In a subsequent randomized, double-blind, placebo-controlled trial, investigators tested the efficacy and safety of hrIL-1ra in the treatment of severe sepsis [16]. Patients (*n* = 696) were randomized to receive standard supportive care and antimicrobial therapy in addition to hrIL-1ra (100 mg) or placebo by IV bolus, followed by a 72-h continuous infusion of hrIL-1ra (2.0 mg/kg/h) or placebo. However, the study was terminated after interim analysis found it unlikely that the primary efficacy endpoints would be met. The 28-day all-cause mortality rate was statistically insignificant between treatment groups (*p* = 0.36). Additionally, the mortality rate did not significantly differ between groups when analyzed based on infection site, infecting microbe, presence of bacteremia, shock, organ dysfunction, or predicted mortality risk at the beginning of the study. Therefore, the authors concluded that a 72-h continuous IV infusion of hrIL-1ra failed to demonstrate a statistically significant reduction in mortality versus standard therapy.

### 2.2. Anti-TNF⍺ Therapy

Tumor necrosis factor alpha (TNFα) is an inflammatory cytokine produced by all cell types but principally is synthesized and released by macrophages and other immune cells in response to infection and noxious stimuli. Many biologics currently on the market treat chronic inflammatory diseases, such as rheumatoid arthritis, by blocking TNFα with targeted antibodies. In sepsis, high levels of proinflammatory cytokines, particularly TNFα have been thought to increase endothelial cell permeability, which can hinder cells from obtaining necessary nutrients [17]. Additionally, the cell’s glycocalyx, the pericellular matrix that surrounds epithelial cell membranes, is susceptible to damage from cytokines, oxidants, and endotoxins [18], further increasing blood-brain-barrier (BBB) permeability [19]. Other deleterious effects of circulating TNFα include myocardial suppression [20,21,22], cellular glutathione depletion [23] and prostaglandin activation [24,25], all of which contribute to MODS.

Nearly 60 trials were conducted to test the efficacy of anti-TNFα antibodies and IL-1 receptor antagonists before the turn of the 21st century. A meta-analysis of the pooled data from these trials yields a small but statistically significant reduction in 28-day all-cause mortality [26]. However, the minor benefits displayed by these treatments were overshadowed by several factors, including variability in trial results, evidence of the potential for harm, and the high financial costs of these agents. Many of these trials suffered from poor design and included patients who would and would not benefit from experimental treatment, thereby confounding the outcome. Never-the-less, enough promise was shown in these early trials to justify additional studies with improved, ‘next-generation’ antibody-based therapies targeting cyto/chemokines. These newer therapies were polyclonal antibody fragments (Fabs), primarily comprised of the variable domain (i.e., light chain), with the rationale being that their smaller size allowed for increased diffusion into extravascular spaces, and a more permissive binding of TNFα.

MONARCS was a phase III RCT performed on over 2500 patients with severe sepsis at 157 centers using the anti-TNFα antibody fragment, afelimomab [27]. This remains one of the largest RCTs ever performed in this patient population. Based on the findings of smaller previous trials using afelimomab [28,29], investigators elected to stratify the initiation of treatment according to patients’ serum IL-6 status. Patients designated as ‘high IL-6’ (≥1000 pg/mL) or ‘low IL-6’ (<1000 pg/mL) were randomly assigned to receive afelimomab or placebo for 3 days and then followed for 28 days. Another factor that differentiated this trial from others was that afelimomab was administered via intermittent IV infusions over 72 h, whereas most other trials only gave a study drug once. A small but statistically significant reduction in 28-day all-cause mortality and organ dysfunction was observed with treatment but only in the ‘high IL-6’ group. Significant decreases in TNF and IL-6 concentrations as well as significant absolute reductions in MODS and SOFA scores were observed vs. placebo in the ‘high IL-6’ group. A high degree of heterogeneity in the inflammatory status of the patients enrolled in the study at baseline was thought, at least in part, to underlie some of the variability in response to therapy. Authors stated they need a more complex statistical model to detect a treatment effect due to the imbalance of inflammatory status in patients at baseline (IL-6 and TNF levels). Like other trials, treatment in MONARCS was initiated within 24 h of a patient meeting inclusion criteria. However, current sepsis treatment guidelines recommend initiating therapy within 3 h of symptom presentation. Thus, additional RCTs with afelimomab where newer guidelines are followed as part of the inclusion criteria may be warranted.

More recently, a double-blind, placebo-controlled, phase IIa trial tested the safety and tolerability of AZD9773, a polyclonal anti-TNFα Fab in patients with severe sepsis [30]. Patients were randomized to receive AZD9773 (including five escalating-dose cohorts) or placebo. For most patients in the AZD9773 cohorts, serum TNF⍺ levels decreased to near-undetectable levels within two hours of administration. In particular, the higher-dose cohorts demonstrated significant lower AUC concentrations of TNFα vs. placebo over the treatment period. This study arguably supports continued evaluation of AZD9773 in larger trials in order to determine whether it yields a clinical benefit. However, the small number of patients enrolled (*n* = 70) and variable baseline TNF⍺ and IL-6 levels were again a confounding factor. A subsequent randomized, double-blind, placebo-controlled phase IIb trial tested the safety and efficacy of two doses of AZD9773 in patients with severe sepsis/septic shock [31]. In this trial, 300 patients were randomized to receive AZD9773 or placebo for five days. There were no significant differences among treatment groups or subgroups stratified by baseline TNFα level, Acute Physiology, Age, Chronic Health Evaluation (APACHE-II) score at screening, infection site, etc., with respect to number of ventilator-free days, mortality rates, or relative risk of death. The authors even noted that these results do not provide justification to further study AZD9773, though certain subgroups, including the low-dose AZD9773 group baseline APACHE II score 21–C25 and >31, did trend toward significance. Despite the ability of AZD9773 to rapidly decrease TNF⍺ levels (reproducing the phase IIa results), this failed to translate to a clinical benefit. However, given the highly variable baseline inflammatory status among sepsis patients, as highlighted from work in the previous trials, it is likely that this trial (*n* = 300) was insufficiently powered to detect a statistically significant difference in clinical parameters.

## 3. Targeting the Innate Immune Response

### 3.1. Alkaline Phosphatase

Alkaline phosphatase (AP) is a ubiquitous protein that has four main isoenzymes with various roles [32]. It has been shown in animal studies that exogenous intestinal AP (IAP) decreases systemic inflammation, particularly detoxification of bacterial lipopolysaccharides (LPS). A study from Malo et al. [33] showed that circulating TNFα and IL-1β inhibit expression of IAP from intestinal endothelial cells, which suggests further susceptibility to bacterial LPS in septic patients due to lack of IAP. A phase II double-blind placebo-controlled clinical trial evaluated whether bovine intestinal alkaline phosphatase (BIAP) improved renal function in 36 patients with severe sepsis or septic shock [34]. A significant improvement of creatinine clearance was observed in the BIAP-treatment group within 24 h, while the placebo group saw a decline of creatinine clearance. Similarly, the BIAP group had significant improvement in average serum creatinine from 1.03 mg/dL to 0.79 mg/dL 7 days after treatment, while placebo group did not show significant improvement at 7 days. Furthermore, this study excluded patients with chronic renal failure on dialysis, while also providing little data on included patients who received renal replacement therapy (RRT). This makes it difficult to extrapolate renal improvement in more severe populations. Although this study showed promise for BIAP to improve renal function in sepsis, the small patient sample size and limited clinical outcome data ultimately weakened the overall impact of the study. However, with a proper study design, it may be possible to discern a small but significant beneficial effect of BIAP in sepsis patients.

### 3.2. Granulocyte-Macrophage Colony Stimulating Factor

Granulocyte-macrophage colony stimulating factor (GM-CSF) is a growth factor that stimulates the immune system by enhancing the host’s cell-killing abilities, principally through activation of neutrophils, macrophages, and monocytes. A preliminary study by Nierhaus et al found that human recombinant GM-CSF upregulated expression of the cell surface antigen mHLA-DR in 9 septic patients with MODS [35]. This is important because low expression of mHLA-DR is a feature of septic patients [36]. GM-CSF was further evaluated in a double-blind, placebo-controlled, RCT of 38 patients with severe sepsis, septic shock, or sepsis-associated immunosuppression [37]. Throughout the study, the treatment group had significantly higher levels of neutrophils, CD4 and CD8 T-lymphocytes, and a significant improvement in APACHE-II score. Despite the treatment arm’s success in many areas, there were multiple important variables that did not reach statistical significance. These include decreased ventilator time, decreased hospital and ICU stay, 28-day mortality, NK and B-lymphocyte counts, etc. This lack of statistical significance in key outcomes was likely due to limitations in the study design (e.g., small patient sample, heterogeneity in presentation, etc.) Additionally, the control group had a tendency toward higher baseline disease severity scores. Inclusion for this trial was broad but mainly consisted of patients that met sepsis and MODS criteria but were relatively healthy otherwise. Details of comorbidities were sparse, and no further stratification occurred, which makes it difficult to ascertain select groups that may or may not benefit with specific acute or chronic conditions. Of note, patients with autoimmune diseases, recent myocardial infarctions, and CPR were excluded. Collectively the study demonstrated that GM-CSF does have therapeutic potential with sepsis, but evidence is still lacking. Here again, larger trials with proper design and sufficient statistical power to assess an effect are needed. 

### 3.3. Intravenous Immunoglobulin 

Intravenous immunoglobulin (IVIG) has been used to treat a number of chronic diseases, including neurological and autoimmune conditions, among others [38]. Therapeutic mechanisms of IVIG is complex and not fully understood, but thought to be multifaceted, including enhanced immune cell function and pathogen clearance, decreased production and/or scavenging of proinflammatory cytokines, and more [39].

There have been many RCTs to determine the efficacy of IVIG as an adjunctive sepsis treatment. Despite these efforts, there are mixed results with no clear consensus. A Cochrane meta-analysis conducted by Alejandra et al. reviewed 43 RCTs and concluded that there was insufficient evidence to support the use of IVIG in septic neonates [40]. Though adult mortality was reduced, the authors stated this benefit was not seen in trials with low risk of bias. Laupland et al. found reduced adult mortality but mention a high level of heterogeneity in the trials analyzed [41]. Additional meta-analyses performed in this timeframe found significant reductions in adult mortality with IVIG use in sepsis patients [42,43]. Interestingly, the majority of these analyses were published between 2004 and 2007, with the exception of Alejandra et al. in 2013, which had come to different conclusions with comparable data available at the time. More definitively, the SBITS study, a large phase III trial, found no difference in mortality rates between the control and treatment groups [44], which conflicts with many of the meta-analyses and supports the findings of Alejandra et al. Of the studies examined, there were subgroups that consisted primarily of comparisons of polyclonal IVIG, IgM-enriched polyclonal IVIG, and monoclonal IVIG between adults and neonates. Currently, there is not enough robust data to suggest any mortality benefit. Until clinical benefit can be proven further, the consensus will likely remain scattered.

Thus, although there is weak evidence to support the use of IVIG for the treatment of sepsis, the lack of agreement and inconsistent results obtained to date suggest that IVIG may only provide a minimal benefit in a select group of patients.

## 4. Targeting the Coagulant Response 

### 4.1. Heparin

Heparin is a commonly used anticoagulant used to treat and prevent thromboembolic events during hypercoagulable states, such as pregnancy, surgery, and myocardial infarctions. During early stages of sepsis, inflammatory mediators interfere and trigger the coagulation cascade, which can lead to organ dysfunction and other complications [45]. Heparin may attenuate coagulation and thereby act to prevent MODS. 

The HETRASE trial was a phase III double-blind, placebo-controlled trial, which aimed to determine if a low-dose continuous infusion of unfractionated heparin (UFH) is an effective complementary treatment in septic patients. The effect of UFH on length of hospital stay, change in MODS score from baseline, and 28-day mortality did not reach statistical significance versus placebo [46]. HETRASE included patients broadly presenting with infection confirmed by both inflammation biomarkers and vital signs. Conversely, admission into the hospital >24 h prior to the study, chronic renal failure, history of organ transplantation, and several others were cause for exclusion. As heparin is safe in renal failure, it may have been beneficial in that population. Despite failing to show clinical improvement, heparin was well tolerated. Thus, ubiquity of access and decent safety profile make heparin a good candidate for future studies as an adjunct sepsis treatment.

### 4.2. Platelet Activating Factor Acetyl Hydrolase 

Platelet-activating factor (PAF) is an endogenous phospholipid that is produced and released by many cell types in the blood, kidneys, liver, skin and other tissues [47]. When PAF is excreted and binds to PAF-receptors, the effect can be proinflammatory in nature, especially in times of infection with concurrent inflammation, such as sepsis. To counter this effect, the PAF-acetylhydrolase (PAF-AH) enzyme is released into plasma where it hydrolyses PAF and other phospholipids [48], thereby inactivating PAF. The idea that PAF-AH may mitigate inflammation was demonstrated in animal models with recombinant PAF-AH [49], and further elucidated in clinical trials [50].

To date, there has been only a single clinical trial assessing recombinant PAF-AH for severe sepsis. The study was a phase III, double-blind, placebo-controlled trial with 2500 participants from 9 countries, in 146 intensive care units (ICU). Unfortunately, the trial was terminated early due to an interim analysis that determined recombinant PAF-AH (rPAF-AH) had limited clinical benefit in the treatment group. Subsequently, the 28-day all-cause mortality, ARDS incidence, ICU-free days, and degree of MODS did not reach statistical significance [51]. Considering the early phase successes and high tolerability, it may be worthwhile to investigate rPAF-AH further as an adjunct agent for sepsis, but for now, it remains a theoretical option.

### 4.3. Activated Protein C 

Protein C is a prothrombin and key anticoagulant factor responsible for maintaining hemostasis and vascular permeability [52,53]. Recombinant human activated protein C (rhAPC) showed compelling dose-dependent reductions in coagulation and lethality in pre-clinical studies of primates with severe sepsis [54]. Based on the success of the early trials, several large RCTs sought to investigate the safety and efficacy of rhAPC in sepsis.

In 2001, the Food and Drug Administration (FDA) approved drotrecogin alfa (rhAPC, Xigris®) for adults with severe sepsis at high risk of death. The FDA, however, also required a study to evaluate the efficacy of drotrecogin alfa for adults with severe sepsis at low risk of death. The ADDRESS trial was a double-blind, placebo-controlled multi-center study that randomized patients to receive drotrecogin alfa or placebo [55]. Unfortunately, trial enrollment was terminated due to a low likelihood of meeting the objective of demonstrating a significant reduction in 28-day all-cause mortality in the treatment group. There were no statistically significant differences in 28-day mortality or in-hospital mortality, even when the patients were stratified by APACHE II score, single vs. multiple organ dysfunction, and use of heparin at baseline. Though a subgroup of patients who were the first enrolled showed significantly higher mortality when treated with drotrecogin alfa vs. placebo. Additionally, the rate of serious bleeding was higher with drotrecogin alfa vs. placebo during the infusion and 28-day study periods. Post hoc analyses of the subgroup of patients who had undergone surgery within 30 days before enrollment with concomitant single-organ dysfunction had significantly higher mortality rates with drotrecogin alfa vs. placebo, which was attributed to sepsis. There were also significantly more bleeding events vs. placebo both during infusion and the 28-day study period. Among those surgical patients who had a bleeding event, more patients in the drotrecogin alfa group died of sepsis-induced multi-organ dysfunction or hemorrhage. The authors concluded that drotrecogin alfa should not be used in patients with severe sepsis at low risk for death (single-organ failure or APACHE II score <25). The number of patients with APACHE II scores >25 was too small to detect a significant mortality effect vs. placebo.

Severe sepsis in pediatric patients is poorly understood compared to severe sepsis in adults. The RESOLVE trial was a randomized, double-blind, placebo-controlled phase III trial that investigated biomarker changes in inflammation and coagulation in pediatrics [56]. Pediatric patients with sepsis-induced cardiovascular and respiratory dysfunction were randomized to receive either drotrecogin alfa or placebo. A significant reduction in D-dimer and thrombin-antithrombin complex was observed in the treatment group compared with placebo. However, a median protein C difference was not observed in the overall treatment population compared with placebo, despite there being a mortality benefit with drotrecogin alfa treatment. In patients >1 year of age, the median percentage change from baseline in protein C was significantly higher with dotrecogin alfa vs. placebo. This difference was not observed in patients <1 year. When stratifying patients with disseminated intravascular coagulation (DIC) by age group, the median percentage change from baseline in protein C activity over time did not differ. However, the authors note that changes in protein C trended toward significant increase in children >1 year of age. In patients without DIC, the change in protein C over time did not differ among age groups. A notable finding from this study was that children displayed a similar response to therapy as adults in the following key areas, (1) severity of coagulation/inflammation and mortality; and (2) pharmacodynamic response to drotrecogin alfa with respect to protein C activity and systemic inflammation.

The ENHANCE group sought to provide further evidence for the safety and efficacy of drotrecogin alfa in severe sepsis [57]. In this single-arm, open-label trial, adults with known/suspected infection, 3–4 SIRS criteria, or at least one sepsis-induced organ dysfunction received i.v. infusion of drotrecogin alfa. Treatment was associated with statistically significant relative and absolute risk reductions (RRR and ARR, respectively) in mortality but a statistically insignificant increased risk for bleeding, notably intracranial hemorrhage. It is important to note that placebo patient mortality was less than or equal to drotrecogin alfa-treated patients when stratified by APACHE II score. Another interesting observation here was that ENHANCE patients treated within 24 h from their first sepsis-induced organ dysfunction had lower mortality vs. those treated after 24 h for all measures of disease severity, except mechanical ventilation and organ dysfunction. Mortality was also numerically higher in all subgroups (when compared to the drotrecogin alfa groups in PROWESS), except for those having ≥4 organs in failure. Additionally, patients ≥65 years of age had significantly higher mortality rates when treated later. Though there is no placebo arm, this study provided supportive evidence for a favorable benefit/risk ratio with rhAPC, and that treatment can be more effective if used sooner.

A prospective, observational, controlled trial tested the efficacy of drotrecogin alfa in the “evolution and outcome” of acute kidney injury (AKI), which complicates acute sepsis-induced cardiopulmonary failure [58]. The authors defined AKI as a persistent oligo-anuria following adequate fluid resuscitation. The most glaring weakness is likely the size of this study, which enrolled only 46 patients. Another major weakness is the lack of blinding since the decision whether to start drotrecogin alfa was left to the discretion of the ICU physician. Oligo-anuria and 28-day mortality were comparable between treatment groups. Baseline characteristics were statistically similar between groups, but this study is far too small to conduct subgroup analyses or conclude that rhAPC is ineffective in treating sepsis-related AKI.

The PROWESS study was a randomized, double-blind, placebo-controlled phase III trial that investigated whether infusion of drotrecogin alfa reduced 28-day all-cause mortality [59]. Patients were randomized to receive either drotrecogin alfa or placebo. Treatment was associated with statistically significant relative and absolute risk reductions (RRR and ARR, respectively) in death but a statistically insignificant increased risk for bleeding. Similar results were found when the groups were stratified by baseline APACHE II score, age, and protein C activity (including the 38 patients who underwent randomization but never received an infusion). Additionally, all subgroup analyses (stratified by APACHE II score, number of dysfunctional organs/systems, sex, age, site of infection, type of infection, and presence/absence of protein C deficiency) yielded similar treatment effects with Xigris® vs. placebo. However, the lack of confirmatory data from other placebo-controlled trials called the results of this trial into question. PROWESS-SHOCK enrolled patients with infection, systemic inflammation, and shock in a randomized, double-blind, placebo-controlled trial to receive drotrecogin alfa or placebo [60]. Drotrecogin alfa did not significantly reduce all-cause mortality at 28 or 90 days versus placebo. This includes all subgroups (recent surgery, number of baseline organ failures, etc.). As a result of these findings, the FDA officially withdrew drotrecogin alfa from the market in October 2011, since no statistically significant decrease in 28-day all-cause mortality was found with treatment in the PROWESS-SHOCK trial.

Alterations in protein C levels appear to correlate with sepsis severity. Therefore, the authors of the RESPOND trial hypothesized that it would be possible to tailor therapy with the use of APC [61]. They also wanted a follow-up to the PROWESS trial that responded to patients’ protein C levels and modified therapy accordingly. This phase II, randomized, double-blind study determined the dose and duration of drotrecogin alfa by measuring protein C levels vs. standard therapy in patients with severe sepsis. Protein C-deficient patients were randomized to standard therapy (drotrecogin alfa 24 mcg/kg/h for 96 h) or alternative therapy (drotrecogin alfa at a higher dose and/or variable duration, 24/30/36 mcg/kg/h for 48–168 h). There was a statistically significant difference in absolute change in protein C from days 1–7 between treatment groups. When patients were stratified by protein C deficiency (moderate vs. severe), alternative treatment demonstrated greater protein C increases (though this was only statistically significant for the moderate protein C deficiency group). Among the moderate deficiency population, shorter infusions (<97 h) resulted in greater mortality. Additionally, higher doses and longer infusions were associated with greater increases in protein C levels with no serious bleeds. The authors concluded that this study successfully proved variable doses and/or durations of drotrecogin alfa can improve protein C levels, a finding which can be incorporated into future studies. However, this trial did not evaluate the efficacy of rhAPC on mortality or other clinical variables, unlike PROWESS and PROWESS-SHOCK. Moreover, the stratification of patients by level of protein C deficiency was done at 24 h rather than baseline, delaying some patients from receiving higher doses, resulting in an imbalance between the severe deficiency alternative and standard therapy subgroups.

## 5. Conclusions/Perspectives

There are several broad consensus findings from these trials that, from our perspective, can be informative to both investigators and clinicians alike. First, it is clear that in all biologic-related RCTs performed to date, administration of recombinant proteins appears to be well-tolerated by the septic patient, even in those experiencing septic shock. When considering that these are critically ill patients, it is somewhat surprising that infusion of a recombinant protein has not been associated with worse outcomes. This is encouraging from a feasibility and safety standpoint. Second, despite the failure of many RCTs in this space, there is more than a little evidence at this point that anti-inflammatory therapies can indeed be beneficial, particularly in select patient populations. Specifically, the complete story for anti-IL-1 and TNFα therapies remains unwritten. In some instances, alterations in study design, even slight alterations, may have led to substantial differences in outcomes. Difficulties in defining proper inclusion/exclusion criteria, and the practical challenges associated with enrolling patients with sepsis remain major obstacles to the field. As observed in the MONARCS trial, it is likely that patients who are still in the ‘pro-inflammatory stage’ of early sepsis could benefit from anti-TNFα treatment. Indeed, it would seem that several of these anti-cytokine therapy RCTs would have benefited from more stringent biomarker-guiding. While IL-6 was used in the baseline stratification criteria for several trials, the rationale for using only this cytokine, rather than multiple cytokines (including the very targets of the therapies themselves) or serum metabolites is not at all clear. Use of serum metabolites such as lactate [62,63] as biomarkers for inclusion in sepsis RCT’s is not routinely employed and yet might be an important variable and offer certain advantages over cytokine measurements. Determination of inflammatory status in a patient can take several hours or longer, due to the time constraints associated with measuring serum cytokines. Given that the newest clinical guidelines for sepsis therapy call for initiation of treatment within the first 3 h of presentation [1,2], this does not seem compatible with contemporary cytokine detection methods (e.g., enzyme-linked immunosorbent assays, ELISAs). Thus, until newer technologies and/or alternative biomarkers emerge, identifying patients who may benefit the most from treatment remains an elusive goal.

Lastly, it is important to recognize the improvements in guidelines for early management of patients with sepsis and septic shock, the product of the “Surviving Sepsis Campaign [2].” These guidelines primarily focus on implementation of broad-spectrum antibiotics, fluid resuscitation, vasodilators, anticoagulation, and other supportive measures, all of which are evidence-based recommendations. This evidence suggests, perhaps, that a major factor in the effective treatment of sepsis/septic shock may not be hidden behind a novel drug target but rather be dependent on controllable factors such as time for pathogen identification, time to initial antimicrobial medication, and recognition of MODS. Unfortunately, not all institutions possess the same instrumentation and expertise, pharmaceutical formularies, type/amount of drugs, equipment, or staffing, which places limitations on what is achievable. Nevertheless, incorporating these new guidelines into the design and implementation of future RCTs in sepsis/septic shock should improve the clarity of outcomes and hopefully lead to beneficial effects on MODS and mortality in this patient population.

## Figures and Tables

**Figure 1 ijms-20-06049-f001:**
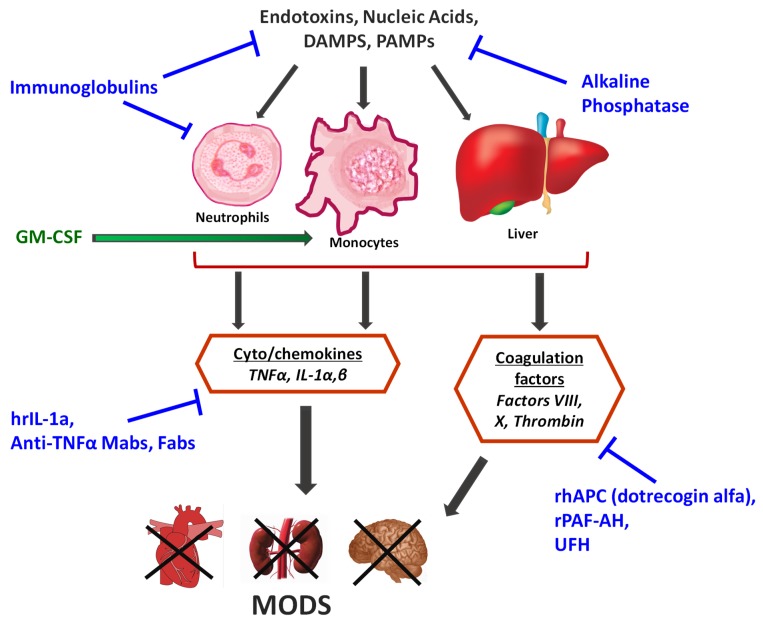
Major sources of inflammation and coagulation in the host response to sepsis that have been targeted in the past 20 years with biologics. Therapies directed at blocking a certain response are shown in blue text, and those that enhance a specific response are in green text.

**Table 1 ijms-20-06049-t001:** Outcomes and Status of RCT’s using Biologics in Sepsis.

Biologic Class	Equivocal Outcomes	Beneficial Outcomes	Current Status/Considerations
rhAPC	[55,56,58,60,64]	[57,59,61]	No clear benefit, discontinued
Heparins	[46,65]		Potential for LMWH, not UFH
hrIL-1a		[14,15,16]	Potential MODS benefit. Need more data in larger trials. More stringent biomarker-guided I/E criteria.
Alk Phos		[34]	Need more data in larger trials.
anti-TNFα	[31]	[27,30]	Potential benefit with Fabs. More stringent biomarker-guided I/E criteria.
IVIG	[44]		No clear benefit.
GM-CSF		[35,37]	Inconclusive. Need more data in larger trials.
rPAF-AH	[51]		Inconclusive. More stringent biomarker-guided I/E criteria.

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
