# Peer review of "Re-Evaluating Biologic Pharmacotherapies That Target the Host Response during Sepsis"

_ijms, 2019, doi:10.3390/ijms20236049_

Round 1
Reviewer 1 Report
Re-evaluating biologic pharmacotherapies that target host response during sepsis
by Tuttle K, McDonald MD, Anderoson EJ
submitted to International Journal of Molecular Science
Dear members of the editorial board, dear authors, thank you for giving the opportunity to review this interesting article submitted by Tuttle and colleagues re-evaluating some biological phamacotherapies targeting host response in sepsis. The paper is well written, but rather descriptive, focusing to failed attempts to combat overwhelming host response in patients undergoing sepsis and suffering from consecutive multiple organ failure. The list of drug is nearly complete, however due to the fact that the authors are following the ‘old school’ concept to cut a maximum level of a compound which is held to be responsible to harm or to substitute a decreased level of another mediators which should be helpful do not broaden our knowledge about the reasons why so many studies failed and there is a unmet medical need for research for so many patients undergoing sepsis, who were treated as yet just supportive and not in any causative manner. One strongly recommended issue for the authors is to describe results of post-hoc analyses which and why some patients groups could profit from the treatment strategy and other subgroups were potentially harmed, thus a result there is no total difference in the mortality rate. For my opinion, there is an urgent need to identify subgroup specific conditions (point of care), to clearly define inclusion criteria, this point is a must to discuss in this review article. Furthermore, I recommend the discussion of biological pharmacotherapies in a reviewing paper only, if
there is a clear subgroup definition which might profit from that therapy there is a clear understanding of underlying pathomechanisms, also to discuss possible adverse effects there is a clear characterization of the host for best performance of personalized therapy.All in all, I cannot recommend acceptance of this article in the submitted version, but the issue needs further consideration, thus I motivate the authors to revise the manuscript and the editors to await this revised version.
Author Response
Authors would like to thank the reviewer for the thoughtful comments and feedback on our review article.
It is undeniable that a major challenge to physicians is to identify and characterize those patients who are most at-risk for organ failure in sepsis, but also those who might benefit or be harmed from biologic immunotherapies such as those discussed in our review. We agree with the reviewer that information concerning this specific challenge was only briefly and superficially discussed in the original draft of the article.
In response to this comment, we have revised our article in many places, providing additional information about patient subgroup analyses in trials involving anti-TNFα, alkaline phosphatase, GM-CSF, IVIG, heparin, and rhAPC (drotrecogin alfa). Specifically, we discuss where evidence suggest harm in a few cases, and also where certain patient subgroups may benefit.
Furthermore, in direct response to the reviewer's comment about stratifying patients by improving the Inclusion criteria, we offer several examples where we believe biomarker guiding can be used, and give details concerning which biomarkers might be of interest. Specifically, we believe that combinations of serum lactate and cytokine levels could be useful during the screening. These details are now briefly discussed in this revision, and notes on these details have been added to the right-hand column of Table 1 (under Current Status/Considerations).
Reviewer 2 Report
The submission from Kristopher Tuttle et al. summarize the main clinical studies on which biological drugs have been tested as pharmacotherapy for multiple organ dysfunction syndrome in sepsi. The review is well written and thorough, however, the authors should better check the manuscript for any typographical error and insert tables that summarize the different targets in a schematic way.
Author Response
Thank you for the evaluation of our manuscript and thoughtful comments for revision. We have carefully proofread our revised article and in response to this reviewer's suggestion, we have revised Table 1 to include more information about the current status of each biologic class, and opportunities where improvements of inclusion/exclusion criteria could lead to successful outcomes.